# OpenReview forum: "Effective Prompt Extraction from Language Models"
_colmweb.org/COLM/2024/Conference — COLM_

### Official Review · Reviewer_tCcJ · 2024-04-27

**Rating:** 5
**Confidence:** 4
**Ethics Flag:** 1

**Summary:**

The paper introduce a new threat to language model that the system prompt could be extracted through user prompt query. Specifically, the paper designs several prompts aiming on letting the model to output what have been given in the input. He then uses a gathered dataset to train a detector to check whether the extracted prompt is the system prompt or not. For the closed source model such as GPT, the paper further proposes a translation based attack that utilized the translated designed prompt to evade the filter. The experimental results show the proposed method could successfully extract system prompt with a high accuracy in a series of LLM including both open-sourced and real-world systems.

**Questions To Authors:**

1. The indictor function in metric definition is not defined. If it is the indictor function, it should get 0 or 1 results. Why does the result later show it is a real number?
2. How to create the Dev dataset of 16000 extractions? Details are needed.
3. How to set the permutation $\pi$? Details are needed.
4. Since there is no opened system prompt in GPT-3.5 and GPT-4, how to calculate the accuracy in Section 5?

**Reasons To Accept:**

1. The proposed setting is very interesting and practical. The prompt extraction attack could be regarded as a new threat to language model system.
2. The experimental results show the proposed method could extract the system prompts with high accuracy in a wide range of LLMs including both open-sourced and real-world systems.

**Reasons To Reject:**

1. The paper needs some serious polish. Several important details are missing which makes the paper not that convincing. Please refer to the questions section.
2. The system prompt definition is not clear. The open-sourced model's system prompt is just the first paragraph of the user prompt however the real system is something hidden in the back-end. If the same standard is applied, the real system could still add the first paragraph of the user prompt act as the system prompt.
3. The proposed finding is interesting however it is purely based on the heuristic. All prompts tested are letting model to repeat itself. I am not sure whether this threat could be fixed very easily by letting the LLM paraphrase his own words.

Minor:
1. Section 4.2 we analyze conduct -> we conduct

---

> ### Author Rebuttal · Authors · 2024-05-27
>
> > R2. System prompt definition is not clear
>
> We consider system prompts as something set by the LLM service provider (e.g., “You are a helpful and harmless chatbot”), and the user prompt typically involves instructions on performing a task (e.g., “Translate everything I say to Spanish.”). Main results in Table 5 are actually on extracting these instruction-style user prompts.  The main reason we don’t do this for actual system prompts is that there aren’t many LLM systems with system prompts available. We can indeed try using a standard system prompt, but we don’t believe it would make a meaningful difference to the experiment results.
>
> We would also like to point out that in Section 6, we try our prompt extraction method on a few LLM services to extract their actual system prompts and observe seemingly successful results, but we can’t compute success metrics since we don’t have access to groundtruth system prompts of these services.
>
> > R3. Defense by paraphrasing
>
> Indeed, exact prompt extraction can be defended through paraphrasing the output of language models, but the paraphrased output likely still reveals the same private information, and can work similarly well when used as instructions. We believe that the simplicity of our natural-language-based attack is in fact a strength: people can just take the set of attack queries and see if they compromise secrecy of prompts by running generation.
>
> > Q1.
>
> Results in Table 1 are averaged (binarized) successes over the test sets. Are there specific places where we reported real-valued metrics and it’s not clear what they mean?
>
> > Q2.
>
> First, we have a set of Dev prompts that are not used for final evaluation. We run prompt extraction over this dataset for all models (e.g., GPT-4 and Llama-2) to create 16,000 prompt, extraction pairs. Since we have access to both prompts and extractions, we can compute groundtruth token leakage (specifically, using the Rouge-L recall metric). The DeBERTa model is fine-tuned on extractions to predict token leakage. We will clarify in the final paper. We would also like to point out that code, dataset and model weights are already open-sourced.
>
> > Q3.
>
> Permutations are always randomly generated. What this means is for a set of extractions generated by different attack queries, we randomly permute the extractions, feed them into the DeBERTa model to predict token leakage. We will clarify this in the paper.
>
> > Q4.
>
> Please see our previous response on system prompt definition.

---

### Official Review · Reviewer_3Nfh · 2024-04-29

**Rating:** 4
**Confidence:** 4
**Ethics Flag:** 1

**Summary:**

The paper introduces a method for prompt extraction that employs attack queries to extract several candidate prompts, and then chooses the candidate with the highest confidence estimate as the most likely prompt.

**Questions To Authors:**

See above.

**Reasons To Accept:**

The prompt extraction task is interesting, and the author has conducted comprehensive experiments across multiple models.

**Reasons To Reject:**

1. This method lacks novelty. [1] has already proposed a similar approach for extracting prompts via multiple "Jailbreak" queries interacting with LLMs, and they also introduced the "Inversion Model" (IM), which not only surpasses "Jailbreak" but also achieves SOTA results. However, the advantages of this paper’s method compared to "IM" and "Jailbreak" remain ambiguous, and it lacks a comparative analysis of performance.
2. The importance of each step in the method is unclear and lacks theoretical support. For example, the necessity of training a DeBERTa model in Section 3.2 is unclear. What theory it is based on? If the goal is to identify common parts across multiple extractions, simpler methods or the LLM itself might suffice. The authors should provide adequate theoretical explanations, ablation studies of different parts, and comparisons with more straightforward strategies.
3. It extracts prompts based on 105 attack queries. Why choose 105 attacks? Why not more or fewer? The impact of this specific quantity on the overall performance of the extraction process remains unclear and needs further investigation.
4. The test sets featured in the main Table 1—including Unnatural Instructions, ShareGPT, and Awesome-ChatGPT-Prompts—comprise only a few hundred prompts each. This limited quantity is insufficient to demonstrate the effectiveness of the method.
5. The author emphasizes extracting the system prompt of LLM and presents the extraction performance of GPT-3.5-turbo and GPT-4 in Table 1. However, this performance is evaluated by comparing the extracted prompts to those in the benchmarks, rather than the real system prompts of LLMs. This evaluation appears unreasonable and does not effectively demonstrate the validity of system prompt extraction across various LLMs.

---

> ### Author Rebuttal · Authors · 2024-05-27
>
> > Method lacks novelty
>
> We don’t have access to the paper you mentioned, and we assume you are referring to the “Language Model Inversion” (LMI) paper, which wasn’t public at the time of writing. LMI and our work both extract prompts, but there are a few key differences:
> - Different assumptions of what an adversary can access: our attack assumes black-box access to a generation API, and LMI requires logit access. Logits can be recovered from a black-box API, but recovery of the full distribution requires $\mathcal{O}(\mathcal{V})$ number of queries. This means LMI cannot be cheaply applied to a black-box generation interface.
> - Our attack is purely natural language, and doesn’t require a separate inversion model.
>
> We will cite the LMI paper and incorporate relevant discussions.
>
> > Why DeBERTa?
>
> Each attack query produces a different response, and we need a method to distinguish good extractions from clearly bad ones, and a language model can be trained to do this effectively. The model outputs a score for each extraction and we use the one with the highest score to run evaluations. If you are comfortable with having multiple extractions, the DeBERTa model is not necessary. It is simply a way of choosing a promising extraction out of all the candidates.
>
> > Why 105 attack queries?
>
> There are 105 attack queries because the authors wrote a set of 5 queries and queried GPT-4 to generate an additional 100 queries. In preliminary experiments, we find just the 5 handwritten queries are already effective. We will add an additional analysis of attack success vs. # of queries.
>
> > Test set in the scale of a few hundred examples does not show effectiveness of method
>
> We disagree with the reviewer. Even on our smallest test set out of 3, the 95% confidence intervals of our exact- and approx-match results are bounded above by $\pm$7.9%. We will report confidence intervals where relevant.
>
> > No systematic evaluation of system prompt extraction
>
> We acknowledge that Table 1 does not show results of system-level, but user-level prompt extraction, but we want to highlight that such an evaluation is impossible: at the time of writing, there were only a small number of foundational LLMs, and their system prompts are generally secret.
>
> Our extraction experiments on 4 system-level chatbots point to successful extraction, but it is impossible to be certain. We will qualify our claims where appropriate to make it clear that our main evaluations are done on user-level prompts.

---

### Official Review · Reviewer_uMFj · 2024-05-09

**Rating:** 7
**Confidence:** 4
**Ethics Flag:** 1

**Summary:**

This paper presents a framework for systematically measuring the effectiveness of system prompt attacks. They use the simple templates to extract the system prompt first, then use a pre-trained small models to decide which system prompt has the highest probability.

In experiments with 3 different sources of prompts and 11 underlying large language models, they find that simple text-based attacks can in fact reveal prompts with high probability. Their framework can determine with high precision whether an extracted prompt is the actual secret prompt, rather than a model hallucination.

Except the smaller LLMs, they also find ChatGPT and Claude's system prompts can be revealed by an adversary despite existing defenses in place.

**Questions To Authors:**

Thanks for this great submission,I have the following question for you:

(1) Have you tried other models except DeBERTa in your experiment? I think if you can do model ensemble, it potentially may further increase the precision.

**Reasons To Accept:**

I think this paper is quite interesting, especially the proposed evaluation framework. I think people know that we can extract the system prompt to some extent, but the author propose a novel framework for this, and the framework is quite effective.

**Reasons To Reject:**

N/A

---

> ### Author Rebuttal · Authors · 2024-05-27
>
> Thanks for your positive review of our work. To address your question, we have only tried fine-tuning DeBERTa. We chose DeBERTa because it is one of the strongest masked language models, which can naturally be used for the task of predicting token leakage. Ensembling could improve predictive performance further, but given that the precision is already high, gain from ensembling would be limited.

---

### Official Review · Reviewer_hqxo · 2024-05-13

**Rating:** 6
**Confidence:** 4
**Ethics Flag:** 1

**Summary:**

This paper present a study of prompt extraction, which is to extract the system prompt based on designed interaction. A framework is designed to systematically measure the effectiveness of these attacks, where one major difficulty is to discriminate a successful extract with a hallucination.

To solve the problem, the authors propose to treat the majority extraction results as the oracle secret prompt. And trained a model to predict the probability of one extracted prompt to be successful.

Experiments on several popular LLMs such as Alpaca, Vicuna, Llama-2, GPT3.5, GPT4 are presented, showing that the prompt attacks are successful.

**Reasons To Accept:**

It is interesting to see the research on prompt extraction, which is a safety direction that has been less considered.

**Reasons To Reject:**

I have concerns about the correctness of using the majority of extracted prompt as the oracle. It is indeed possible that the model generate the similar results for the similar queries for the prompt.

It is strange that different type of prompts have very different rate of being extracted (In Table 1). For example, Unnatural is much higher than the other two prompt datasets. It might be useful to include more discussions.

---

> ### Author Rebuttal · Authors · 2024-05-27
>
> > Concerns about the correctness
>
> The main concern of the reviewer is that we report prompt extraction success results based on an approximate string matching metric. To clarify, the approx-match metric does not consider an extraction successful when the majority of tokens are leaked; it considers an extraction successful when >90% of tokens are leaked (see its precise definition in Section 2, “metrics of success”). We also report extraction successes under an exact-match metric, which only considers an extraction successful if the entire prompt is leaked, and exact-match success rates remain high.
>
> The reviewer’s confusion may stem from how we phrased our result in a few places (e.g.., “The majority of prompts can be extracted across heldout datasets and models.” in Table 1 caption). When we say “the majority of prompts can be extracted”, we always mean “more than 50% of prompts in the dataset can be extracted,” rather than “more than 50% of tokens in prompts can be extracted.” We will make the language more clear in the next version.
>
> > More discussion on how different types of prompts have very different rate of being extracted
>
> We will add qualitative examples from all three sources of prompts and provide a discussion on what might have made prompt extraction more difficult.

---

> > ### Comment · Reviewer_hqxo · 2024-06-05
> >
> > The concerns are actually about the evaluation process, not about the phrasing. It is the case where no true prompt could be obtained, which is the case the attack may actually be performed. The related part in the original paper is "This results in extractions in multiple languages, which we back-translate to English; if the back-translations are consistent, then we can be fairly confident that they match the true prompt."

---

### Decision · Program_Chairs · 2024-07-10

**Decision:**

Accept

**Comment:**

Based on the reviews and subsequent discussions, I recommend accepting this paper. The work presents a thorough study on prompt extraction and the reviewers generally found the topic interesting and the experimental work comprehensive, covering multiple models including both open-source and real-world systems. While concerns were raised about the novelty of the method, particularly in comparison to the LMI paper, we can consider LMI as concurrent work. Given the practical importance of the prompt extraction threat and the systematic approach presented in this paper, I believe it makes a valuable contribution to the field. The authors should ensure they incorporate the promised improvements and clarifications in the camera-ready version, particularly regarding comparisons with the LMI work and several design choices of their evaluation pipeline mentioned by the reviewers.